# Effect of Thermal Aging on Mechanical Properties and Color Difference of Glass Fiber/Polyetherimide (GF/PEI) Composites

**DOI:** 10.3390/polym14010067

**Published:** 2021-12-24

**Authors:** You Song, Jiangang Deng, Zhuolin Xu, Yu Nie, Zhenbo Lan

**Affiliations:** 1Wuhan Nari Limited Liability Company of State Grid Electric Power Research Institute, Wuhan 430074, China; l17720559398@163.com (Y.S.); dengkelvin@163.com (J.D.); xuzhuolinwh@139.com (Z.X.); whnr18327057305@163.com (Y.N.); 2State Grid Electric Power Research Institute, Nanjing 210000, China

**Keywords:** glass fiber reinforced polyetherimide, thermal aging, mechanical property, color difference

## Abstract

This research study is aimed at evaluating the mechanical characteristics in terms of tensile strength and flexural strength of glass fiber reinforced Polyetherimide (GF/PEI) under different thermal aging. Tensile testing and bending testing were performed on the thermally aged polyetherimide composites. The mechanical properties of the thermally aged samples were also correlated with their color difference. The experimental results showed that both the tensile strength and flexural strength of the GF/PEI composite samples decreased with increasing aging temperature. However, the elastic modulus of the composite samples is nearly independent on the thermal aging. The thermally aged samples exhibited brittle fracture, resulting in low strength and low ductility. The loss in strength after thermal aging could be also linked to the change of their color difference, which can indirectly reflect the change of the strength for the composites after thermal aging.

## 1. Introduction

Polyetherimide (PEI) is an amorphous engineering thermoplastic that exhibits high temperature resistance, outstanding mechanical and electrical properties [1,2,3]. This high-performance polymer also exhibits high flame resistance and low smoke generation, which makes it an ideal material of choice in internal components of microwave ovens, electrical and electronic products, and automotive, appliance, and aerospace, and transportation applications [4,5]. It is a convenient material for the study of structure-property relationships and in particular their dependence on thermal history. It exhibits strong relaxations, associated with the glass transition, which makes it important in comparing dynamical mechanical and dielectric spectra.

Researchers have investigated novel production processes to increase the properties of PEI in specific applications, such as the addition of alumina or silica nano-particles which increase the ultimate strength of PEI, or the fabrication of nanofoams with higher specific modulus and thermal resistances than other foams [6,7]. Today, fiber-reinforced polymer composites, thanks to their potential advantages such as high specific strength, stiffness and being lightweight, have taken the place of traditional metals in many areas, such as automotive, aircraft industry, wind turbine, and marine applications. Despite the many advantages of polymer matrix composites compared to conventional metals, the biggest challenge is to ensure that they work with minimal external inspection and maintenance, especially over the years. The durability of polymer matrix composite materials under environmental conditions has become an ongoing concern for the industry and researchers due to increasing use of polymer composite materials. In some application areas, environmental factors, such as heat, moisture, ultraviolet rays and loads may degrade the material properties.

In order to improve the physical, thermal, and mechanical properties, PEI composites reinforced by fiber or filled by mineral filler have been developed to increase tensile strength, flex modulus, creep resistance, impact resistance, dimensional stability, heat, and chemical resistance [8,9,10]. The aim of combining fibers and resins that are different in nature is to take advantage of the distinctive material features of either component to result in an engineered material with desired overall composite action for specific applications. The fibers are added to increase the load-carrying capability of the composite material. They have extremely high tensile strength and exhibit tensile properties along the fiber’s length, and the fibers have a generally higher stress capacity than ordinary steel and are linear-elastic until failure. The main function of the matrix is to provide support for the fibers and assist the fibers in carrying the loads. It also distributes the loads evenly between fibers such that all the fibers are subjected to the same amount of strain. The matrix provides stability to the composite material and acts as a binding agent in a structural component in which the fibers are embedded. It also holds the fibers together and protects the fibers from the environment. The choice of the fibers greatly affects the composite performance. The most common reinforcement for the polymer matrix composites is a glass fiber. The glass fibers are divided into three classes: E-glass, S-glass, and C-glass. The most commonly used is E glass, which is used for repair and strengthening of structures. Glass fiber reinforced PEI (GF/PEI) composites are the most successful commercialization type among all kind of modified PEI materials and are widely used in aerospace, automobiles, electronics, chemical machinery due to its excellent mechanical properties, heat resistance, flame retardancy, chemical resistance and attractive cost benefit [11,12,13,14]. GF/PEI composites are being considered for use in different fields of application, ranging from sporting goods to structural materials for automotive, aerospace, and power industries, where the long-term properties are of primary importance. In such applications, the composites are exposed to harsh and variable environments ranging from variations in temperature and exposure to moisture, including elevated temperature immersion, hot exposures and electric shock [15]. 

Thermal aging behavior of composites is of special interest because of their expanding use for structural applications where increased temperatures are common environmental conditions. Sometimes a sudden increase in temperature may be quickly followed by a sudden decrease in the temperature. There are significant chemical and structural changes in the fiber-reinforced PEI–matrix composite, especially in the matrix networks, which take place during thermal aging. Delamination and micro-cracking are some of the most frequently observed damaging phenomena that may develop in PEI composites exposed to high temperatures conditions. It is important to understand the aging mechanism of PEI composites for their use in thermal aging environments. The mechanical behavior of composites depends on the ability of the interface to transfer stress from the matrix to the reinforcement fiber. The development of a comprehensive understanding of the mechanisms of aging and environmental exposure-related deterioration, for the purposes of prediction of service-life and durability, is of immense importance to extend the service life of such systems [16,17,18,19]. Aging is generally divided into two classes, physical and chemical. Physical aging is the type of reversible aging involving both property and dimensional changes. On the other hand, in the case of chemical aging, there is irreversible degradation of the molecular structure caused by mechanisms such as chain scission, changes in crosslinking density and oxidation. In service, failures of glass–fiber-reinforced polymeric composites are commonly attributed to aging of the material in its particular environment, brought about by a combination of the effects of heat, water, and mechanical stresses on the material. Several studies have shown the important effects of absorbed water and aging temperature on the physical and mechanical properties of composite materials [20,21,22,23,24,25].

Yilmaz studied the effects of hydrothermal aging on GF/PEI composites [26], and reported that the physical properties like glass transition temperature and mechanical properties deteriorate because of water absorption of epoxy polymer matrix composites. Vina et al. [27,28] studied the effects of natural and accelerated aging on the static and dynamic behavior of GF/PEI and reported that GF/PEI composites show excellent behavior after natural and accelerated aging from the point of view of their mechanical properties. Murmu et al. [29] investigated the effect of graphite powder addition on the wear behavior of GF/PEI composite, they found that the addition of graphite powder could improve the flexural strength and wear resistance of the composites. Arici et al. [30] investigated the effect of hydrothermal aging on both continuous carbon and glass fiber reinforced PEI composites, and found the plasticization for hydrothermally aged glass fiber reinforced PEI composite and the higher thermal expansion mismatch problem between carbon fiber and PEI matrix caused the composite to decrease remarkably in static energy values. Literature studies prove that the mechanical properties deteriorate because of microcrack formation due to uneven thermal contraction and expansion at hydrothermal conditions [31,32]. Therefore, there is a need to retain the physical and mechanical properties of fiber-reinforced polymer composites in the thermal environment to enhance the durability and reliability of the advanced material. In this context, the experimental determination of their mechanical properties over a wide range of temperatures, as well as quantification of the effect of aging on the mechanical properties, are extremely important for design and in-service monitoring purposes [33]. 

Therefore, in this present work, the effects of thermal aging on the mechanical behavior of glass fiber reinforced PEI were studied. Thermal aging treatments at different temperatures from 80 to 145 °C were conducted on GF/PEI composite plates and, the mechanical properties such as tensile properties and bending properties were evaluated. Besides, the change of color difference of GF/PEI composites which was exposed to different thermal aging temperatures were also correlated with their mechanical properties.

## 2. Materials and Methods

### 2.1. Materials and Sample Preparation

The materials used in this work were composites with a matrix of PEI reinforced with 20% glass fibers. Its matrix is a semi-crystalline high-performance thermoplastic with high thermal stability, low water absorption and high mechanical resistance and stiffness. For this study, PEI powder was supplied by Honghe Limited Corporation (Zigong, China). The GF (FR5301B-2000) was supplied by Chongqing Polymer Composite International. Plate samples and dog-bone samples were supplied by a local company in the form of injected plates with a barrel set temperature of 300 °C and a tool temperature of 100 °C. Both the plate samples and the dog-bone shaped samples were prepared, and used to investigate the effect of thermal aging on the mechanical properties and color change after thermal aging at different temperatures.

### 2.2. Thermal Aging Procedure

Thermal aging treatments of the plate composite samples was performed under air in ovens (HZ-2004, Lyxyan, Dongguan, China) at different temperatures: 85 °C, 100 °C, 115 °C, 130 °C and 145 °C for 180 h. The samples were put in the furnace when the temperature rises to the required temperature and remained constant. For comparison, the original plate samples without thermal aging were also prepared. These samples are referred to as GF/PEI-x samples, where x denotes the aging temperature applied.

### 2.3. Mechanical Testing

Tensile properties were evaluated with the dog-bone shaped specimens according to ASTM D3039 standard. The tensile test was carried out using a mechanical testing machine (MTS E45, MTS, Eden Prairie, MN, USA) at room temperature with a constant crosshead speed of 2 mm/min. An extensometer of 50 mm in length was employed to measure the displacement in the gauge length region. The average values were recorded after testing five specimens of each sample. For tensile testing, the tensile test specimens, which were of dimensions 2 mm × 10 mm × 150 mm, were prepared from composite plates. Tensile strength, tensile strain, and elastic modulus of the samples were measured and reported as average values.

The bending performance of the plate samples was tested in three-point bending mode according to ASTM D7264 using a universal testing machine (MTS E45) with a crosshead speed of 1 mm/min. The flexural test specimen’s dimensions are 80 mm × 10 mm × 4 mm and support span is 64 mm, as shown in Figure 1b. Similarly, in tensile tests, each test was repeated five times. The flexural strength of the composites was calculated by Equation (1)
*σ* = 3*PL*/2*bd*^2^(1)
where *L*, *b*, *d*, and *P* are the support span, specimen width, specimen depth, and flexural load, respectively.

### 2.4. Fracture Surface Observation

The samples for fracture surface observation were cut from the tensile fractured samples. The fractured surfaces of the samples were coated with carbon before observation. Fracture surface observations were carried out using a field emission scanning electron microscopy (FE-SEM TESCAN MIRA III, Brno, Czech Republic) operated at an accelerating voltage of 5 kV. 

### 2.5. Color Characterization

The color change of the thermally aged samples was tested in colorimeter (x-rite Color-Eye 7000A, Gretag Macbeth, Grand Rapids, MI, USA) and the Lab value was used to evaluate the color change of materials. In which, the *L* value presents luminance, ranging from 0 to 100, indicating from dark to bright. The *a* value ranges from −128 to 128 indicates from green to red. The *b* value ranges from −128 to 128 indicates from blue to yellow. Usually, the color change can be characterized by Δ*E* calculated as Equation (2):Δ*E* = (Δ*L*^2^ + Δ*a*^2^ +Δ*b*^2^)^1/2^(2)
at meanwhile, the grayscale and glossiness (*G60*) values of the composite samples were also measured five times for averaged values.

## 3. Results and Discussion

### 3.1. Mechanical Behavior

Tensile strength is one of the most important properties for indicating the mechanical performance of materials and it is described as the ability of a material to resist a force that tends to pull it apart. The tensile stress–strain curves of the plate samples are given in Figure 1a. The stress–strain curves of the samples can be divided into two parts: (1) the first part is linear and represents the elastic deformation, as shown by the inset in Figure 1a; (2) the second part is nonlinear and represents the elastic–plastic deformation. Tensile strength values of GF/PEI-0, GF/PEI-85, GF/PEI-100, GF/PEI-115, GF/PEI-130 and GF/PEI-145 were found as, 73.7, 73.3, 69.9, 67.2, 66.1 and 58.2 MPa, respectively. These results show that, the tensile strength values of the composite samples gradually decrease with increasing aging temperature, especially when the aging temperature exceeds 100 °C.

The tensile strain values of the composite materials are given in Figure 1c. The tensile strain value of the composite composites gradually decreases from 4.6 ± 0.5% to 3.3 ± 0.3% with increasing the aging temperature.

The elastic modulus values of the composite materials are given in Figure 1d. It is clearly observed that, however, the elastic modulus seems independent on the aging temperature, all the samples exhibited an elastic modulus of approximately 2 GPa, as shown in Figure 1d. The thermal aging has a negligible effect on the elastic modulus of the GF/PEI samples.

Figure 2a shows the typical stress-displacement curves of the three-point bending test of the composite specimens after different thermal aging. The displacement was measured at the position where the load was applied. For all the specimens, the stress-displacement curves increased linearly until the ultimate load where the specimen fractured. It is clearly observed that the stress-displacement responses of the aged specimens significantly decreased compared to that of the original specimen, which indicates the deterioration in the fracture toughness of the aged specimens compared to the original specimen. Figure 2b reveals that the bending strength decreases with increasing the aging temperature, which is similar to the results of the tensile strength, as shown in Figure 1b. With increasing the aging temperature, the flexural strain also reduces, as shown in Figure 2a. Therefore, both the flexural strength and strain decrease with increasing thermal aging temperature. This is due to degradation of polymer matrix under high temperature aging, which leads to the brittleness of the composites. This may be related with the temperature increase leading to matrix swelling and microcrack formation at the interface.

### 3.2. Microscopy Investigation

Figure 3 shows the total tensile fracture surfaces of the aged specimens. The original sample exhibits a rough fracture surface with multiple cracks, as shown in Figure 4a. It is observed that overall, the fracture surfaces of the samples after thermal aging show brittle features, which are indicated by initiation, transition and final fracture regions. The initiation region surrounds the origin of the crack and is associated with slow crack propagation. The fracture surface is relatively featureless, exhibiting a textured microflow. The transition region is characterized by a steady increase in the surface roughness and the beginning of river lines. In the final fracture region, the crack grows rapidly, and the conical-shaped patterns develop. The parts of the figure symbolized by the numbers 1, 2 and 3, respectively, indicate the initiation, transition and the final fracture regions. The thermally aged samples, however, exhibit relatively flat fracture surfaces. The river lines are clearly observed in the transition region, as shown in Figure 3.

Figure 4 presents the fracture morphologies of the composite samples, GF/PEI-0 and GF/PEI-85 (aged at 85 °C) and GF/PEI-145 (aged at 145 °C). It can be seen that the fiber-matrix adhesion seems to be of very high quality for the original composites (before aging), as shown in Figure 4a,b. A thin layer of PEI resin coated on the surface of glass fibers was observed, also indicating a good interfacial adhesion between resin matrix and glass fiber. For the GF/PEI-85 composite, a few PEI resin remained on the fiber surface, as shown in Figure 4c,d. However, the smooth surface of glass fibers is observed in the GF/PEI-145 composite, as shown in Figure 4e,f. It indicates that there is obvious separation between glass fiber and matrix in the GF/PEI composites aged at high temperature. Therefore, the reduction of tensile strength of aged GF/PEI composites is attributed to the decreased interfacial adhesion between resin matrix and glass fiber due to the degradation of PEI resin. Thermal aging has destroyed the matrix and leads to the separation of matrix and fibers, which in return reduces the mechanical retention of PEI original material.

### 3.3. Color Difference

Table 1 shows the color difference of the aged GF/PEI samples after thermal aging at different temperatures. It can be observed from Table 1 that the color of the GF/PEI composite samples did not change significantly, except that the ΔE value changed from 1.27 to 3.02 with increasing the aging temperature, as shown in Table 1. The gray scale of the aged samples exhibits no significant change as compared to the original sample. This may be a reason that the holding temperatures used were quite lower than their glass transition temperature. However, the gloss (G60) of the samples decreases gradually from 90.7 to approximately 70 with increasing the aging temperature, as shown in Table 1. The difference of the color change indicates that the GF/PEI composite samples is not easily oxidized during the aging process. This result coincides with the change of mechanical property, namely, the mechanical properties decrease accompanied with the decrease of ΔE and G60 values, as shown in Figure 5. As a result, the correlation between mechanical properties and color difference of the GF/PEI composite samples indicates that the strength of the composite materials after thermal aging is predictable by using color change analysis, which is very suitable for actual application of the composites.

## 4. Conclusions

This present work studied the effects of thermal aging on mechanical responses of PEI plate composites that are reinforced with glass fibers. Tensile and three-point tests were conducted. Experimental findings indicate that the thermal aging in GF/PEI composite samples appears to be responsible for variations in mechanical properties. Both the tensile strength and the flexural strength of the GF/PEI composite samples decreased with increasing the aging temperature from 80 to 145 °C, which can be also reflected by the change of their color difference. The ductility of the samples also decreased with increasing the aging temperature. The elastic modulus of the samples did not change significantly with aging temperature, indicating that it is almost independent on the thermal aging. Microscopy observations using SEM works indicate that the fractured surface of thermally aged composites shows a brittle manner without significant plastic deformations, indicating that the GF/PEI composite samples is obviously degraded by thermal aging at the present temperature range. Thermal aging of GF/PEI composites is a major concern for structural durability in high-temperature service environments. This work is expected to be helpful in understanding the effects of thermal aging on the durability of GF/PEI composites.

## Figures and Tables

**Figure 1 polymers-14-00067-f001:**
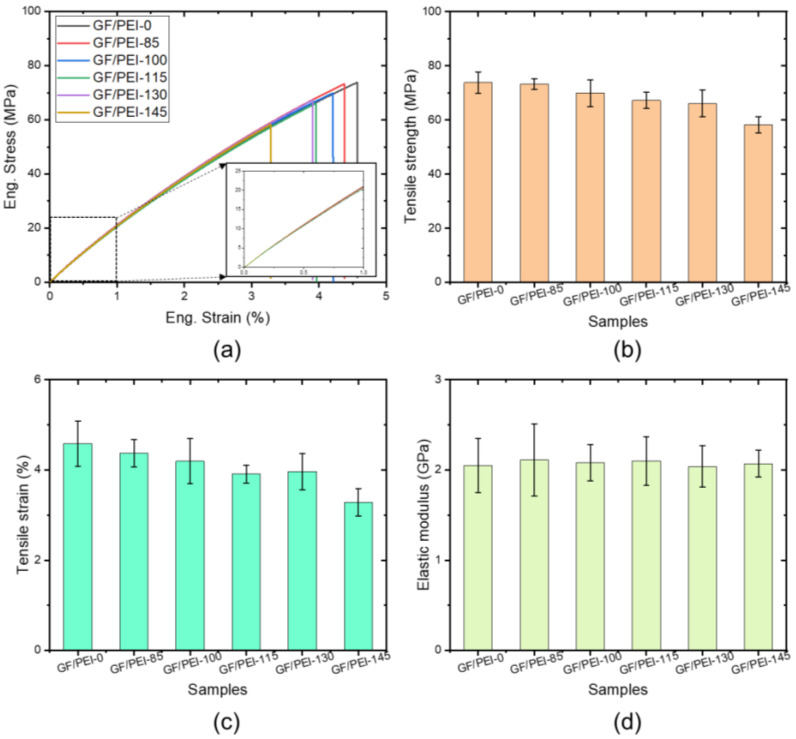
Typical tensile stress-strain curves of the aged specimens (**a**) and tensile strength (**b**), tensile strain (**c**) and elastic modulus (**d**) as a function of temperature.

**Figure 2 polymers-14-00067-f002:**
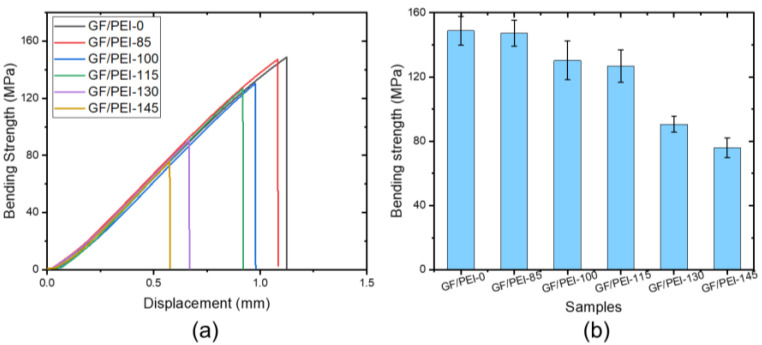
Bending stress-displacement curves of the samples (**a**), and bending strength of the aged samples as a function of temperature (**b**).

**Figure 3 polymers-14-00067-f003:**
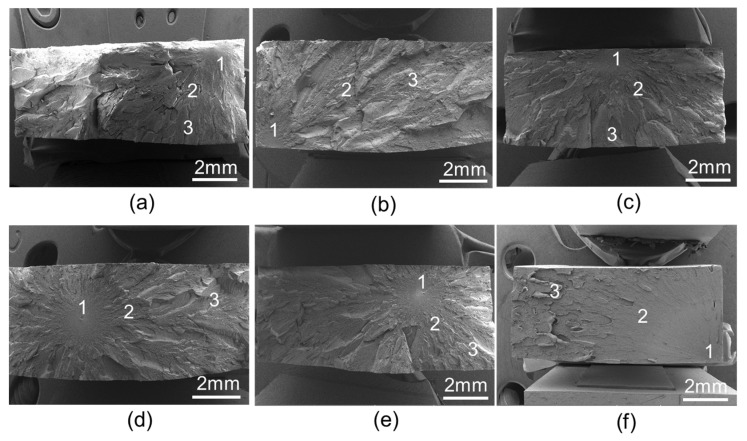
Micrographs of taken from the fractured surface of the aged samples. (**a**) GF/PEI-0, (**b**) GF/PEI-85, (**c**) GF/PEI-100, (**d**) GF/PEI-115, (**e**) GF/PEI-130, (**f**) GF/PEI-145.

**Figure 4 polymers-14-00067-f004:**
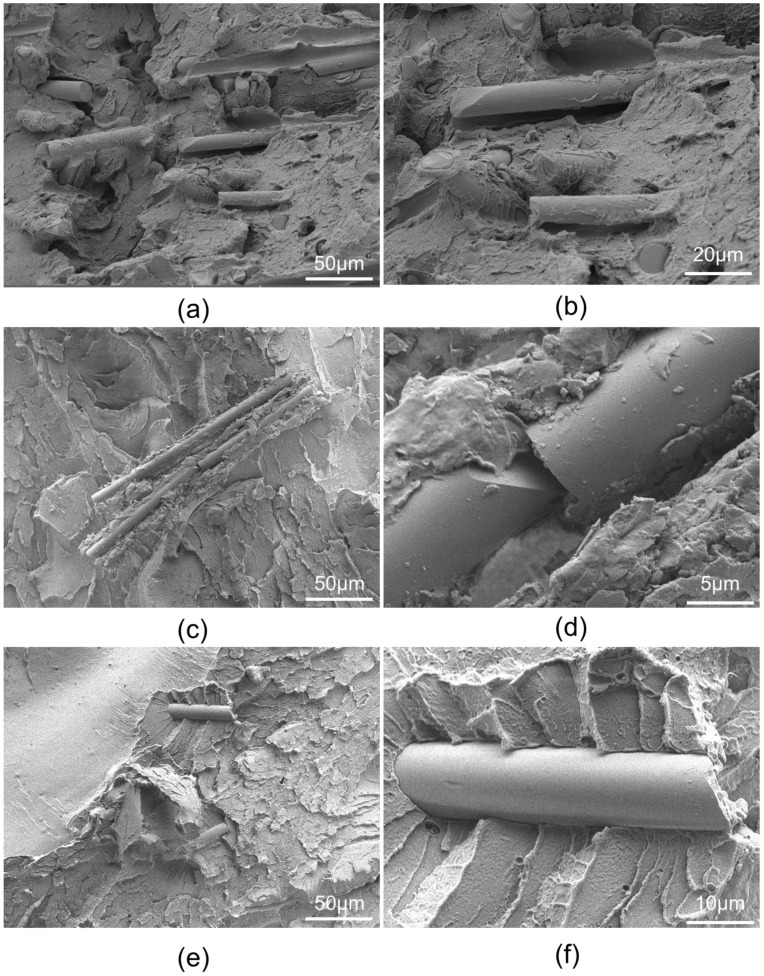
High-magnification images of the fracture surfaces: (**a**,**b**) GF/PEI-0 and (**c**,**d**) GF/PEI-85 and (**e**,**f**) GF/PEI-145.

**Figure 5 polymers-14-00067-f005:**
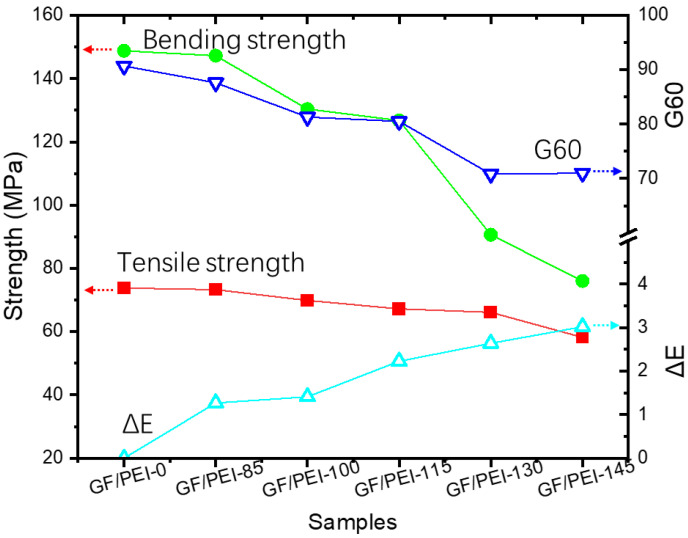
Correlation between mechanical strength and color difference of the GF/PEI samples.

**Table 1 polymers-14-00067-t001:** Color difference of the aged GF/PEI composites.

Temperature (°C)	L	a	b	ΔE	Gray	G60
Original	10.24	−0.32	−0.16	0	28 ± 2	90.7 ± 4.8
85	9.48	−1.34	−0.14	1.27 ± 0.25	32 ± 3	87.6 ± 6.7
100	9.18	−1.17	−0.52	1.41 ± 0.31	25 ± 2	81.3 ± 5.2
115	12.46	−0.56	−0.04	2.23 ± 0.52	25 ± 2	80.5 ± 4.3
130	12.83	0.19	−0.25	2.64 ± 0.32	33 ± 3	70.8 ± 3.9
145	12.58	−1.2	1.54	3.02 ± 0.25	32 ± 3	71.0 ± 4.2

## Data Availability

All data used during the study appear in the submitted article.

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
