# Peer review of "Effect of Thermal Aging on Mechanical Properties and Color Difference of Glass Fiber/Polyetherimide (GF/PEI) Composites"

_polymers, 2021, doi:10.3390/polym14010067_

Round 1

Reviewer 1 Report

In this work, authors discuss the impact of thermal aging on the mechanical performance of glass fiber reinforced Polyetherimide (GF/PEI), suggesting an indirect relationship between the variation of the mechanical properties of the materials to their color difference. There are some really interesting aspects discussed in this well written manuscript, however some points need to be addressed in order to be recommended for publication.

  1. In the experimental section, there is little to no information regarding the fabrication process of the examined systems. Authors report in the introduction, and rightly so, that the manufacturing parameters play a vital role in the final properties of the composites.
  2. Besides the actual value of temperature, the variation of mechanical properties under thermal aging is heavily dependent on the time the materials spent under the respective thermal load and/or the number of thermal cycles. Authors should include these parameters in the experimental section. What is the influence of time and number of thermal cycles in the mechanical performance of the examined systems?
  3. Is there a rationale behind the selection of the specific temperatures employed in this work? What are the working conditions of these systems? Have the authors investigated the thermal stability and glass transition temperature of the samples? How are they affected by thermal aging?
  4. Authors state that the results regarding the color change were taken after examining 5 samples. However, standard deviation for ΔE and G60 values are not listed in the respective table. Finally, a comment regarding the variation of the grayscale should be added to the manuscript.

Reviewer 2 Report

Dear author,

The effects of thermal aging on mechanical properties and color difference change of GF/PEI composites were studied experimentally. Some positive results and some practical reference values have been obtained. The topic of the article is also consistent with the aims & scope of Polymers.

Here are some suggestions for the author's reference.

  1. In lines 125-131, it is suggested to give the specific manufacturers, specifications, and performance parameters of PEI and glass fiber, and the suppliers of flat and dog bone samples for testing, as well as their molding process parameters and equipment model specifications and manufacturers.
  2. Please describe in detail the thermal aging operation process, such as whether the sample is put in before the furnace temperature rises or is put in after the furnace temperature rises to the required temperature and remains constant, as well as the specification, model, manufacturer and origin of the heating furnace, etc.
  3. In lines 139-152 of the mechanical properties test, it is recommended to list the test equipment specifications, manufacturer, place of origin, etc., as well as the thickness dimensions of tensile test specimens.
  4. In the size of the dog bone sample in Figure 1 (a), how can the lengths of two 60mm sizes be so different in the same figure? Is it correct that the width size of both ends of the sample is 60mm?
  5. In section morphology observation and color difference characterization, it is recommended to list the manufacturer and place of origin of the corresponding equipment.
  6. The difference between figure (b) and figure (d) in Figure 5 is not obvious, and even the interface performance shown in figure (b) is worse than that shown in figure (d). It is suggested that the photo of (b) be replaced with the photo of single glass fiber covered more and closer by PEI in figure (a), while there are also many such glass fibers in figure (a), such as the glass fiber sticking out from the middle of the picture.
  7. In all results and discussions, it is recommended that some discussion be made about the causes or mechanisms associated with them, as discussed in the stress-displacement section.

Reviewer 3 Report

The present manuscript aims at studying the effects of thermal aging on the mechanical behavior of glass fiber reinforced PEI. In my opinion, there is no novelty is this study.  Anyway, I recommend to Editor to accept the manuscript after revision.

My comments are:

  1. The Authors should state better the novelty of the manuscript.
  2. Mechanical behavior: Include the standard of tensile test.
  3. Figure 1 is not necessary.
  4. “Tensile strain is a measure of the deformation of an object under tensile stress and is defined as the fractional change of the object's length when the object experiences tensile stress.” And “Elastic modulus refers to the hardness of a material and the materials with high elastic modulus also have high hardness.’’ - The above sentences are confusing and unnecessary.
  5. “The tensile strain value of the composite composites gradually decreases from 4.6% to 3.3% with increasing the aging temperature”. - Including error in the analysis in the above sentence.
  6. “Tensile strength values of GF/PEI-0, GF/PEI-85, GF/PEI-100, GF/PEI-115, GF/PEI-130 and GF/PEI-145 were found as, 7, 73.3, 69.9, 67.2, 66.1 and 58.2 MPa, respectively, as given in Figure 2b.”  I recommend to the Authors to decide how the results are presented. (Writen or Figure, not both)

Round 2

Reviewer 1 Report

I am now able to recommend this work for publications, since the authors have addressed all the major issues.